# Estimation of Daily Average Shortwave Solar Radiation under Clear-Sky Conditions by the Spatial Downscaling and Temporal Extrapolation of Satellite Products in Mountainous Areas

**Yanli Zhang** [1,2,*] and **Linhong Chen** [1,3]

1   College of Geography and Environment Science, Northwest Normal University, Lanzhou 730070, China;
    2018212329@nwnu.edu.cn
2   Key Laboratory of Resource Environment and Sustainable Development of Oasis, Lanzhou 730070, China
3   Yunfu Urban and Rural Planning Research Center, Yunfu 527300, China
*   Correspondence: zyl0322@nwnu.edu.cn; Tel.: +86-13893198303

**Abstract:** The downward surface shortwave radiation (DSSR) received by an inclined surface can be estimated accurately based on the mountain radiation transfer model by using the digital elevation model (DEM) and high-resolution optical remote sensing images. However, it is still challenging to obtain the high-resolution daily average DSSR affected by the atmosphere and local topography in mountain areas. In this study, the spatial downscaling and temporal extrapolation methods were explored separately to estimate the high-resolution daily average DSSR under clear-sky conditions based on Himawari-8, Sentinel-2 satellite radiation products and DEM data. The upper and middle reaches of the Heihe River Basin (UM-HRB) and the Laohugou area of Qilian Mountain (LGH) were used as the study areas because there are many ground observation stations in the UM-HRB that are convenient for DSSR spatial downscaling studies and the high-resolution instantaneous DSSR datasets published for the LHG are helpful for DSSR temporal extrapolation studies. The verification results show that both methods of spatial downscaling and temporal extrapolation can effectively estimate the daily average DSSR. A total of 3002 measurements from six observation sites showed that the 50 m downscaled results of the Himawari-8 10-min 5 km radiation products had quite a high correlation with the ground-based measurements from the UM-HRB. The coefficient of determination ($R^2$) exceeded 0.96. The mean bias error (MBE) and the root-mean-squared error (RMSE) were about 41.57 W/m$^2$ (or 8.22%) and 49.25 W/m$^2$ (or 9.73%), respectively. The fifty-two measurements from two stations in the LHG indicated that the temporal extrapolated results of the Sentinel-2 10 m instantaneous DSSR datasets published previously performed well, giving $R^2$, MBE, and RMSE values of 0.65, 41.06 W/m$^2$ (or 7.89%) and 88.90 W/m$^2$ (or 17.07%), respectively. By comparing the estimation results of the two methods in the LHG, it was found that although the temporal extrapolation method of instantaneous high-resolution radiation products can more finely describe the spatial heterogeneity of solar radiation in complex terrain areas, the overall accuracy is lower than that achieved with the spatial downscaling approach.

**Keywords:** daily average downward surface shortwave radiation; spatial downscaling; temporal extrapolation; Himawari-8; Sentinel-2; DEM

## 1. Introduction

The local daily average of the downward surface shortwave radiation (DSSR) is an essential parameter for many models such as surface energy balance, climate monitoring, quantitative remote sensing inversion, and glacier/snow ablation [1–6]. DSSR reflects irradiance received by a given surface, and it can be measured through empirical or physical models based on meteorological observation elements (such as sunshine duration, temperature, relative humidity, etc.) from ground sites [7–10]. Since the 1960s, satellite remote sensing has gradually become an important data source for the estimation of solar

shortwave radiation because it can accurately detect atmospheric and surface features. In particular, the estimated solar radiation by remote sensing provides spatial distribution characteristics compared with the ground-based observations of DSSR. Nowadays, both polar-orbiting satellites and geostationary satellites can provide shortwave solar radiation products with various temporal and spatial resolutions. However, in mountainous areas, DSSR is simultaneously affected by atmospheric attenuation such as water vapor and aerosols, as well as terrain effects such as the slope inclination, aspect, obstruction coefficient and sky view factor, which make it is still challenging to obtain daily average DSSR estimations with a high spatial resolution [11]. The obstruction coefficient indicates whether the surface is sunlit, and the sky view factor is the fraction of the sky dome seen by the surface. Currently, there are two methods that can be used to estimate high-resolution daily average DSSR: one is to perform spatial downscaling on geostationary satellite radiation products with a high temporal resolution; the other is to perform temporal extrapolation on polar-orbiting satellite radiation products with a high spatial resolution.

Polar-orbiting satellites such as Terra/MODIS, Aqua/MODIS, and VIIRS can retrieve 1 km DSSR due to their high spatial and spectral resolutions based on a high-resolution digital elevation model (DEM) [12]. However, these satellite radiation products have a limited temporal resolution in that they can only provide the instantaneous DSSR at the satellite's overpass time. Consequently, numerous studies have made great efforts to extend the estimated instantaneous solar irradiance to the daily average DSSR using three main methods: the interpolation method [12,13], the Meteorological Radiation Model (MRM) [14,15] and the sinusoidal model simulation method [16–23]. Some studies have substituted daytime variations in atmospheric parameters into empirical or physical radiative transfer models, extended the instantaneous radiation to any moment, and obtained the daily average value through integration [12,13]. Some researchers have combined atmospheric parameters such as the aerosol optical depth and the single scattering albedo retrieved by polar-orbiting satellites and the meteorological data such as relative humidity and atmospheric pressure from field meteorological stations to estimate the daily DSSR using the MRM [14,15]. Unfortunately, while these methods boast a high level of accuracy in local areas, they lack general applicability to different regions due to a lack of adequate ground meteorological measurements. Since the plotted variation in daily DSSR variation is consistent with the shape of a downward parabola, some researchers have also employed quadratic polynomial regression to estimate the daily average DSSR from Terra/MODIS and Aqua/MODIS products [16]. Sinusoidal and piecewise sinusoidal models have been widely used to extend one or two MODIS observations to daily values since the daytime radiation variation is similar to a sinusoidal curve [17–23]. Yan et al. [23] compared three methods of estimating the daily average DSSR—the quadratic polynomial regression, the piecewise sinusoidal, and the sinusoidal model based on the MODIS instantaneous DSSR—and verified that the sinusoidal curve can more precisely describe the diurnal variation in shortwave solar radiation at ground observation sites.

Compared with polar-orbiting satellites, geostationary satellites have unique advantages in terms of obtaining the diurnal variation characteristics of DSSR due to their high temporal resolution (10-min to hourly) [24–26]. However, traditional geostationary meteorological satellite products have relatively low spatial and spectral resolutions, making it difficult to accurately obtain the contents of the main atmospheric components, such as clouds, aerosols, and water vapor, which strongly attenuate surface irradiance and exhibit rapid diurnal changes at a regional scale. Therefore, some researchers have performed spatial downscaling on geostationary satellite radiation products to obtain high spatial resolution DSSR [27,28]. Such methods are generally divided into empirical methods and physical model methods. Empirical spatial downscaling methods mainly include interpolation methods, such as weighted average interpolation and Kriging interpolation, which are quite effective in flat areas, but their reliability rapidly decreases as the terrain complexity increases [29,30]. Based on a high-resolution DEM, many studies have tried to improve the DSSR estimation accuracy by considering various topographic factors in the process of

spatial downscaling. Ruiz-Arias et al. [27] downscaled 5 km MSG radiation products to 90 m by calculating the obstruction coefficient and sky-view factor based on a 90 m DEM. Haurant et al. [31] downscaled the 10 km EUMETSAT (European Organization for the Exploitation of Meteorological Satellites) radiation data to 90 m based on their elevation, obstruction coefficient, and sky-view factor. Bessafi et al. [32] increased the spatial resolution of the radiation product of 0.05° CM-SAF to 250 m for Reunion Island with the help of the obstruction coefficient and sky-view factors.

Although the above-mentioned empirical downscaling methods for DSSR improve the spatial heterogeneity of surface irradiance, they still ignore the interaction between solar radiation and the atmosphere. The physical downscaling approach takes into account the effects of both topography and atmospheric parameters on DSSR. Wang et al. [28] downscaled 15 min MSG satellite radiation products with a spatial resolution of 3 km to 30 m by adding topographic parameters (slope, aspect, topographic shading, sky-view factor) and atmospheric parameters (relative optical air mass, Rayleigh optical thickness, Linke turbidity factor) based on the ALOS 30 m DEM in a complex terrain area of the northern Iberian Peninsula. The results showed a high correlation with ground measurements taken from the BSRN (Baseline Surface Radiation Network) site [33], and the determination coefficient was 0.97.

In recent years, a variety of advanced products, including polar-orbit satellites and geostationary satellites, have been released for free. Compared with MODIS and Landsat TM, the MSI (Multispectral Imager) images loaded on Sentinel-2A/B (S2) satellites launched by the European Space Agency (ESA) in June 2015 and March 2017 can provide higher radiation resolution (12 bit), higher spatial resolution (10 m), which can invert more reliable atmospheric and surface parameters. Zhang et al. [34] accurately estimated 10-m DSSR at the S2 A/B overpass time between September 2017 and August 2018 based on S2 atmospheric and surface reflectance products in Laohugou Glacier No. 12 of the Qilian Mountains in China. Himawari-8 (H8), a new-generation geostationary meteorological satellite launched by the Japan Meteorological Agency on 7 October 2014 acquired full-disk observations in 16 multispectral bands (3 visible, 3 near-infrared, and 10 infrared bands). H8 shows great promise for monitoring aerosols and clouds for the accurate estimation of DSSR [35]. The H8 DSSR products have attracted considerable attention because of their high temporal resolution (2.5–10 min) and high spatial resolution (0.5–2 km).

The purpose of this paper is to, respectively, introduce the Himawari-8 and Sentinel-2 satellite radiation products into spatial downscaling and temporal extrapolation methods to estimate the daily average DSSR under clear-sky conditions in mountain areas. The remainder of this paper is organized as follows: Section 2 introduces the data used in this study; Section 3 describes the estimation methods for the daily average DSSR; the estimated results and discussion are presented in Sections 4 and 5, respectively; and the conclusions are drawn in Section 6.

## 2. Study Area and Data

### 2.1. Study Area

As shown in Figure 1, the study area includes two parts: the upper and middle reaches (UM-HRB) of the Heihe River Basin and the Laohugou area (LHG) of the Qilian Mountains. The Heihe River Basin, spread over $14.3 \times 10^4$ km$^2$, is the second-largest inland river basin in the arid region of Northwest China [36]. The UM-HRB has different topographic conditions and different climatic regions with altitude differences greater than 4000 m. There are six observation sites distributed throughout this area, which allowed us to further explore the effects of the spatial downscaling method.

The LHG is located on the northern slope of the western end of the Qilian Mountains in Subei Mongolian Autonomous County, Gansu Province. This region is located at a high altitude, has low temperatures throughout the year, and has well-developed continental glaciers [37]. The shortwave solar radiation is of great significance to glacial melting and

retreat. The DSSR estimated by Zhang et al. [34] at the overpass of the S2 satellite in this area was used to calculate the daily average DSSR using the temporal extrapolation method.

**Figure 1.** Overview of the study area: (**a**) the LHG of Qilian Mountain; (**b**) the UM-HRB.

*2.2. Data*

Four kinds of datasets were used in this study: 10 min H8 radiation products, 10 m DSSR datasets at the S2 A/B satellite overpass time, and 30 m DEM and ground-based measurements, as shown in Table 1. Pyranometer data collected from observation sites were used for model validation.

**Table 1.** Details of the datasets used from the two study areas.

| Datasets | Spatial Resolution | Temporal Resolution | Date | Region | Amount of Data | Applications |
|---|---|---|---|---|---|---|
| S2 DSSR | 10 m | 2–5 days | 1 September 2017–25 August 2018 | LHG | 52 scenes | Temporal extrapolation |
| H8 product | 5 km | 10 min | 13 January 2018–29 September 2019 | UM-HRB LHG | 531 scenes 84 scenes | Spatial downscaling |
| DEM | 30 m | \ | \ | UMHRB/LHG | 1 scene | Terrain factors |
| Pyranometer | \ | 10 min | 13 January 2018–29 September 2019 | UM-HRB | \ | Precision verification |
| | \ | 30 min | 1 September 2017–25 August 2018 | LHG | \ | |

Table 2 presents basic information about the radiation observation sites in the two study areas, including six ground stations in the UM-HRB and two ground observation sites in the LHG.

**Table 2.** Basic information about the pyranometer observation sites in two study areas.

| Region | Station | Lon (°) | Lat (°) | Alt (m) | Slope (°) | Aspect (°) | Number of Measurement Points |
|---|---|---|---|---|---|---|---|
| UM-HRB | Zhangye (ZY) | 100.45 | 38.98 | 1460 | 5.40 | 64.29 | 531 |
| | Huazhaizi (HZZ) | 100.32 | 38.77 | 1731 | 2.68 | 90.00 | |
| | Arou (AR) | 100.46 | 38.05 | 3033 | 1.59 | 18.44 | |
| | Heihe remote sensing (HRS) | 100.48 | 38.83 | 1560 | 2.15 | 51.34 | |

**Table 2.** *Cont.*

| Region | Station | Lon (°) | Lat (°) | 50 m DEM | | | Number of Measurement Points |
|---|---|---|---|---|---|---|---|
| | | | | Alt (m) | Slope (°) | Aspect (°) | |
| | Dayekou (DYK) | 100.29 | 38.56 | 2703 | 7.40 | 284.35 | 409 |
| | Daman (DM) | 100.37 | 38.86 | 1556 | 1.66 | 135.00 | 469 |
| LHG | AWS2 (4550 m) | 96.54 | 39.48 | 4549 | 7.78 | 30.96 | 42 |
| | AWS1 (5040 m) | 96.56 | 39.43 | 5028 | 6.35 | 0.00 | 37 |

### 2.2.1. Himawari-8 Radiation Products

The area observed by H8 ranges from 60°S to 60°N and from 80°E to 160°W. The 10 min radiation products with a spatial resolution of 5 km were chosen for this study. To investigate the applicability of the spatial downscaling method in UM-HRB with complex terrain, the 531 scenes of H8 radiation products collected from January 2018 to September 2019 were selected in combination with available ground observations. Moreover, 84 scenes of H8 radiation products in the LHG were selected to further explore the effect of this method in the glacier area. The H8 radiation products used in this study were provided by the Japan Aerospace Agency (JAXA) (https://www.eorc.jaxa.jp/ptree/, (accessed on 1 June 2022)). A detailed description of the H8 products is available in the article [38].

### 2.2.2. Sentinel-2 Instantaneous DSSR

Sentinel-2 provides new opportunities for shortwave solar radiation estimation at the regional or local scale because of its high spectral resolution and spatial resolution. The instantaneous 10-m DSSR data from the S2 satellite used in this study were obtained in our previously published study through a mountainous radiation transmission model in the LHG [34]. By comparing 52 in situ observations under clear sky conditions, it was found that the estimated shortwave solar radiation data at the transit time of the satellite were accurate with an MBE of $-16.0$ W/m$^2$ and an RMSE of 73.60 W/m$^2$. The detailed algorithm of the mountain radiation transmission model can be found in Zhang et al. [11,34].

### 2.2.3. Digital Elevation Model

The DEM is used to obtain basic auxiliary data for shortwave solar radiation estimations in mountainous areas. The DEM datasets of two study areas were obtained for free from the National Tibet Plateau Data Center (http://data.tpdc.ac.cn/zh-hans/, (accessed on 1 June 2022)). These were extracted with the Advanced Spaceborne Thermal Emission and Reflection Radiation Global Digital Elevation Model (ASTER-GDEM). This dataset has a spatial resolution of 30 m, a vertical accuracy of 20 m, and a horizontal accuracy of 30 m [38].

### 2.2.4. In-Situ Measurements

As shown in Figure 1 and Table 2, the ground observations were obtained from six sites in the UM-HRB and two automatic weather stations (AWSs) on Laohugou Glacier No. 12 in the LHG. The stations in the UM-HRB have different surface topographic features, and the temporal resolution of the data was 10 min. The elevations of the two stations in the LHG are 4550 m (AWS2) and 5040 m (AWS1), respectively, and the DSSR ground-based measurements were made every 30 s.

## 3. Methods

To estimate the daily average shortwave solar radiation in mountain areas, two methods based on DEMs were explored: a spatial downscaling method based on the H8 10-min radiation products and a temporal extrapolation method based on S2 instantaneous DSSR under relatively clear-sky days, as shown in Figure 2.

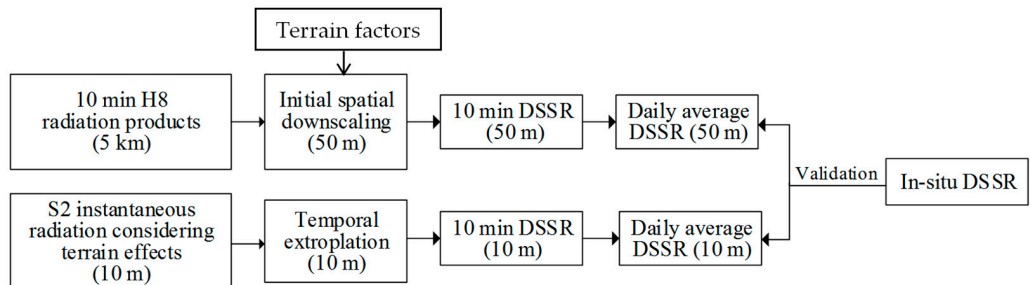

**Figure 2.** Flowchart of the daily average DSSR estimation.

*3.1. Spatial Downscaling*

The spatial downscaling of Himawari-8 radiation products included three key steps [28]: (1) the 5 km H8 radiation product (10-min) was primitively downscaled into 50 m irradiance components on a horizontal surface; (2) each component of downscaled DSSR mentioned above was separately subjected to topographic correction to obtain the more realistic DSSR at the inclined surface based on the DEM; (3) the time-integrated method was applied to obtain high-resolution daily average DSSR by integrating 10-min radiation data into a day.

Generally, the DSSR received on a surface consists of direct, diffuse, and surrounding-reflected irradiance, of which the direct irradiance $E_b$ and diffuse irradiance $E_d$ are the main components. Because the calculation of surrounding-reflected irradiance is complicated, it is ignored in the process of spatial downscaling. Thus the DSSR can be expressed as follows:

$$DSSR = E_b + E_d = E_0 \cdot cos\ \theta \cdot (T_b + T_d) \tag{1}$$

where $E_0$ is the solar irradiance at the top of the atmosphere (TOA), which is derived from the solar constant and the correction coefficient of the Sun–Earth distance; $\theta$ is the solar zenith angle; and $T_b$ and $T_d$ are the direct and diffuse transmittance, respectively.

Since geostationary satellites with shortened revisit times (around 10 min), provide in detail the atmospheric conditions, the actual atmospheric transmittance can be retrieved from the literature [28]. In the initial spatial downscaling, the DSSR value of the coarse-resolution pixel ($DSSR_c$) is assumed to be the DSSR average value of all its corresponding high-resolution pixels, which is obtained from the H8 5-km radiation products. Thus, the initial downscaled DSSR at the high-resolution pixel scale (50 m) on a horizontal surface can be estimated as follows:

$$DSSR_{h,i} = DSSR_c \cdot \frac{n cos\theta_{h,i} \cdot (T_{bh,i} + T_{dh,i})}{\sum_{i=0}^{n} cos\theta_{h,i} \cdot (T_{bh,i} + T_{dh,i})} \tag{2}$$

where $DSSR_{h,i}$ is the DSSR of the *i*-th high-resolution pixel within the corresponding coarse pixel after the spatial downscaling. $\theta_{h,i}$, $T_{bh,i}$ and $T_{dh,i}$ denote the solar zenith angle and the direct and diffuse transmittance of the *i*-th pixel and can be calculated according to the simple parameterized empirical formula presented by Wang et al. [28] by considering the Linke turbidity factor, relative optical air mass, Rayleigh optical thickness, and surface elevation. Therefore, the two components of the downscaled irradiance can be calculated by the following formulas:

$$E_{bh,i} = E_0 \cdot cos\theta_{h,i} \cdot T'_{bh,i} \tag{3}$$

$$E_{dh,i} = E_0 \cdot cos\theta_{h,i} \cdot T'_{dh,i} \tag{4}$$

where $E_{bh,i}$ and $E_{dh,i}$ are the direct and diffuse irradiance downscaled by the high-resolution pixel, and $\theta_{h,i}$ is the corresponding solar zenith angle. $T'_{bh,i}$ and $T'_{dh,i}$ are the actual transmittances of the direct and diffuse irradiance corrected by the initial downscaled DSSR with the high-resolution pixel. The detailed calculation formulas can be found in the literature [28].

Secondly, based on the terrain factors, such as the slope, aspect, sky-view factor and obstruction coefficient, topographic correction was applied to the spatial downscaling results of the 50-m DSSR estimated above by

$$E'_{bh} = E_{bh} \cdot V_s \cdot cos\varphi_h / cos\theta_h \tag{5}$$

$$E'_{dh} = E_{dh} \cdot V_{iso} \tag{6}$$

where $E'_{bh}$ and $E'_{dh}$ are the two downscaled components of the direct and diffuse irradiance, respectively. $V_{iso}$ and $V_s$ are the sky-view factor and the obstruction coefficient, which were both calculated by the Relief Visualization Toolbox (RVT) developed by Zakšek et al. [39]. $\varphi_h$ is the local solar illumination angle on a sloped grid, which was determined by the solar zenith and azimuth angles, slope, and aspect of the sloped pixel. Finally, the daily average DSSR was obtained by integrating the 10-min downscaled shortwave solar radiation during the daytime.

### 3.2. Temporal Extrapolation

The sinusoidal model proposed by Bisht et al. (2005) was adopted to simulate the diurnal variations of the DSSR with single instantaneous shortwave solar radiation data points estimated from the satellite on clear-sky days, as follows:

$$DSSR(t) = DSSR_{max} sin\left[\left(\frac{t - t_{rise}}{t_{set} - t_{rise}}\right)\pi\right] \tag{7}$$

$$DSSR_{max} = DSSR_{ovp} / \pi sin\left[\left(\frac{t_{ovp} - t_{rise}}{t_{set} - t_{rise}}\right)\pi\right] \tag{8}$$

$$DSSR_{avg} = \int_{t_{rise}}^{t_{set}} DSSR(t)dt / \int_{t_{rise}}^{t_{set}} dt = 2DSSR_{ovp} / \pi sin\left[\left(\frac{t_{ovp} - t_{rise}}{t_{set} - t_{rise}}\right)\pi\right] \tag{9}$$

where *DSSR(t)* represents the shortwave solar radiation at a given time *t*, and $DSSR_{max}$ is the maximum DSSR during the day. $t_{rise}$ and $t_{set}$ are the local sunrise and sunset times, which were calculated by the local latitude and date without considering topographic effects. $t_{ovp}$ indicates the satellite overpass time, and $DSSR_{ovp}$ is the instantaneous DSSR at the satellite overpass time.

Obviously, the key to obtaining the daily average DSSR is to accurately estimate the instantaneous DSSR received by the slope pixel at the S2 satellite overpass before using temporal extrapolation of the sinusoidal model. In this study, the $DSSR_{ovp}$ datasets (10 m) were taken from the results of our published paper [34] collected from the LHG, which were estimated based on a mountain radiative transfer scheme by combing the Li mountain radiation algorithm [40] with the Yang broadband atmospheric attenuation model [41]. These DSSR datasets performed very well, and the details of the algorithm principle and estimation steps with the help of DEM and S2 L2A products are given in the literature [34].

## 4. Results

### 4.1. Evaluation against Ground-Based Measurements

Several statistical parameters were used to validate the estimated results with ground-based measurements, including the mean bias error (MBE), the mean bias error percentage (MBE%, the MBE divided by the mean observation), the root mean square error (RMSE), the root mean square error percentage (RMSE%, the RMSE divided by the mean observation), and the coefficient of determination ($R^2$), and the corresponding formulas are detailed in Huang et al. [42]. Furthermore, the high-resolution DSSR obtained by the spatial downscaling method represents radiation in the direction of the ground surface normal vector, while the measurements of the ground station radiometer are horizontally positioned. Therefore, for effective verification, a simple cosine correction was carried out on the measurements of the station according to the solar elevation angle, surface slope, and aspect [28].

### 4.1.1. Evaluation of the Original H8 Radiation Products

Research conducted by Zhang et al. [34] proved that the instantaneous DSSR estimated by the S2 satellite at the S2 satellite overpass is high accuracy (MBE = $-16.0$ W/m$^2$; RMSE = 73.60 W/m$^2$) based on the mountain radiative transfer scheme. To validate the accuracy of the original H8 10-min radiation products in the study area, 10,473 in situ measurements taken over 27 days (15 days in 2018, 12 days in 2019) were selected from six ground stations under almost-clear-sky conditions throughout the day in the UM-HRB. Figure 3 shows that the values of the original H8 10-min products are consistent with the ground observations, the overall accuracy is relatively high (R$^2$ = 0.95, RMSE = 84.85 W/m$^2$, and MBE = 50.40 W/m$^2$), and the bias comes mainly from clouds, aerosols and bright albedo [38].

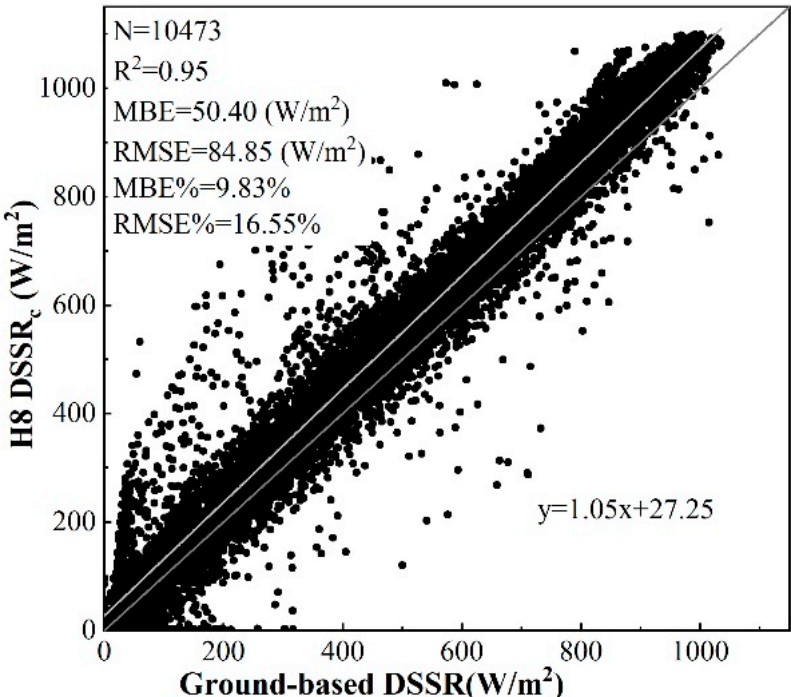

**Figure 3.** Scatterplots of the original H8 10-min DSSR products vs. the corresponding ground-based measurements in the UM-HRB.

### 4.1.2. Evaluation of the Spatial Downscaling

To verify the estimation accuracy of the daily average DSSR by spatial downscaling, the accuracy of the downscaled H8 10-min products was first evaluated. Based on the pyranometer data from the 27 selected days, 3002 observations collected over 8 days with the best clear-sky conditions were selected for quantitative statistical analysis, as shown in Table 3. It can be seen from the accuracy statistics that the spatial downscaling of shortwave solar radiation is highly correlated with ground observations: the $R^2$ value exceeds 0.96, and the RMSE is 69 W/m$^2$ (13.37%), and MBE is 40.95 W/m$^2$ (7.93%). The experimental results show that, compared with the original H8 10-min products, the downscaled radiation products are more reliable.

**Table 3.** Accuracy comparison between the downscaled and original H8 10-min products in the UM-HRB.

| Datasets | MBE (W/m$^2$) | MBE% | RMSE (W/m$^2$) | RMSE% | R$^2$ | Number of Measurement Points |
|---|---|---|---|---|---|---|
| H8 DSSR$_h$ | 40.95 | 7.93 | 69.00 | 13.37 | 0.96 | |
| H8 DSSR$_c$ | 45.85 | 8.88 | 71.29 | 13.81 | 0.97 | 3002 |

By integrating the H8 10-min downscaled DSSR, the daily average shortwave solar radiation was determined. Similarly, the 10-min ground observations from the eight clear-sky days mentioned above were also integrated into the daily average DSSR for accuracy validation. However, due to the lack of measurements from the DYK station, 45 observations from six stations were selected for statistical analysis in the UM-HRB. Figure 4 illustrates the scatterplots and statistical results, showing that the downscaled 50-m daily average DSSR is in good agreement with the field measurements with an $R^2$ value of 0.92. The downscaled algorithm was shown to precisely estimate the DSSR received on the ground surface, and the value of the RMSE (49.25 W/m$^2$) was smaller than the daily average RMSE estimated by Bisht et al. [17]. However, the results indicate that, overall, the spatial downscaling method has a degree of overestimation (MBE = 41.57 W/m$^2$), which is related to the original accuracy of H8 radiation products [43–45].

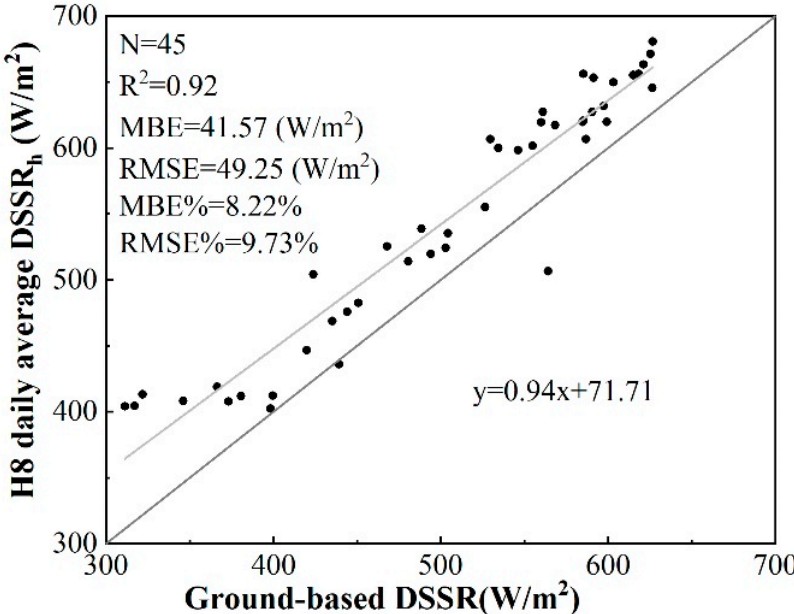

**Figure 4.** Scatterplots of the downscaled daily average DSSR vs. the corresponding ground-based measurements in the UM-HRB.

### 4.1.3. Evaluation of Temporal Extrapolation

After choosing the instantaneous DSSR datasets published previously for 62 clear-sky days at the S2 satellite overpass time from September 2017 to August 2018 in the LHG [34], the sinusoidal model was applied to extend the daily average DSSR. Due to the difficulty of obtaining in situ observations from two AWS stations on Laohugou Glacier No. 12, only 52 ground-based measurements of the daily average DSSR were selected to verify the estimated DSSR by the sinusoidal model.

As shown in Figure 5, the sinusoidal extrapolation method for the daily average DSSR performed well based on the instantaneous DSSR at the S2 satellite overpass time with an MBE of 41.06 W/m$^2$. The daily average estimated DSSR values were consistent with the ground measurements ($R^2$ = 0.65). Although the RMSE (88.90 W/m$^2$, 17.02%) was relatively large, this value cannot reflect the distribution of solar radiation estimation accuracy, because DSSR values were overestimated in high regions and underestimated in low regions. Further research found that the main atmospheric parameters, such as atmospheric water vapor and aerosol, vary rapidly in the valley glaciers, and there are fewer completely clear skies throughout the day, which gives the sinusoidal model a certain level of uncertainty. However, in any case, for mountain glaciers where ground observations are very difficult, this level of estimation accuracy is acceptable.

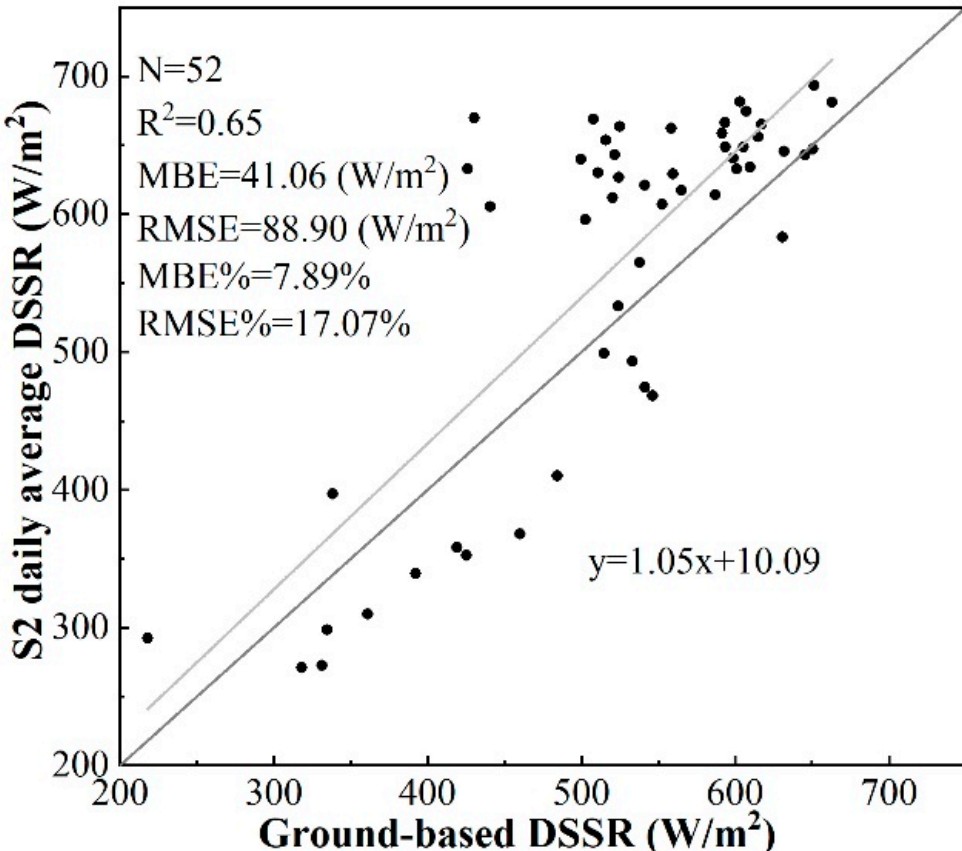

**Figure 5.** Scatterplots of the daily average DSSR estimated by the sinusoidal model vs. the corresponding ground-based measurements obtained from the LHG.

### 4.2. Mapping of DSSR

#### 4.2.1. Mapping of Downscaled DSSR

Based on the DEM and H8 10-min radiation products, 531 downscaled daily average DSSR measurements from 13 January 2017 and 29 September 2019 were simulated. Compared with the original H8 product, downscaled DSSR can not only improve the simulation accuracy of surface irradiance, but can also provide more detailed information on spatial distribution characteristics. To facilitate the analysis of the spatiotemporal distribution of downscaled DSSR, we selected data from three typical moments (9:00, 13:00, and 17:00 h) on 25 July 2018, a sunny day for analysis. As can be seen from Figure 6a–c, the spatial distribution on the H8 radiation product tended to be smooth, and it is difficult to see the DSSR variation with the fluctuation of the surface. However, it can be seen from Figure 6d–f that the spatial heterogeneity of DSSR at the 50 m scale is extremely different from that of the original H8 product, and the value of the downscaled DSSR varies with the terrain. This is because terrain correction is carried out in the process of spatial downscaling, and the effects of local solar illumination angle, obstruction coefficient, and sky-view factors are considered.

As shown in Figure 6g–i, the irradiance on the slope pixel has high spatiotemporal variation characteristics. Generally, a slope facing the sun, namely, the sunny slope, can receive more solar radiation energy, while the shadowy slope covered by the mountain receives less radiation energy. Furthermore, the irradiance received by a given surface varies greatly at different times of the day. The above results strongly confirm the reliability of the spatial downscaled DSSR for describing the terrain effects. In addition, affected by the cloud cover, there is an obvious low value (blue area) in the southwest of Figure 6c at 17:00.

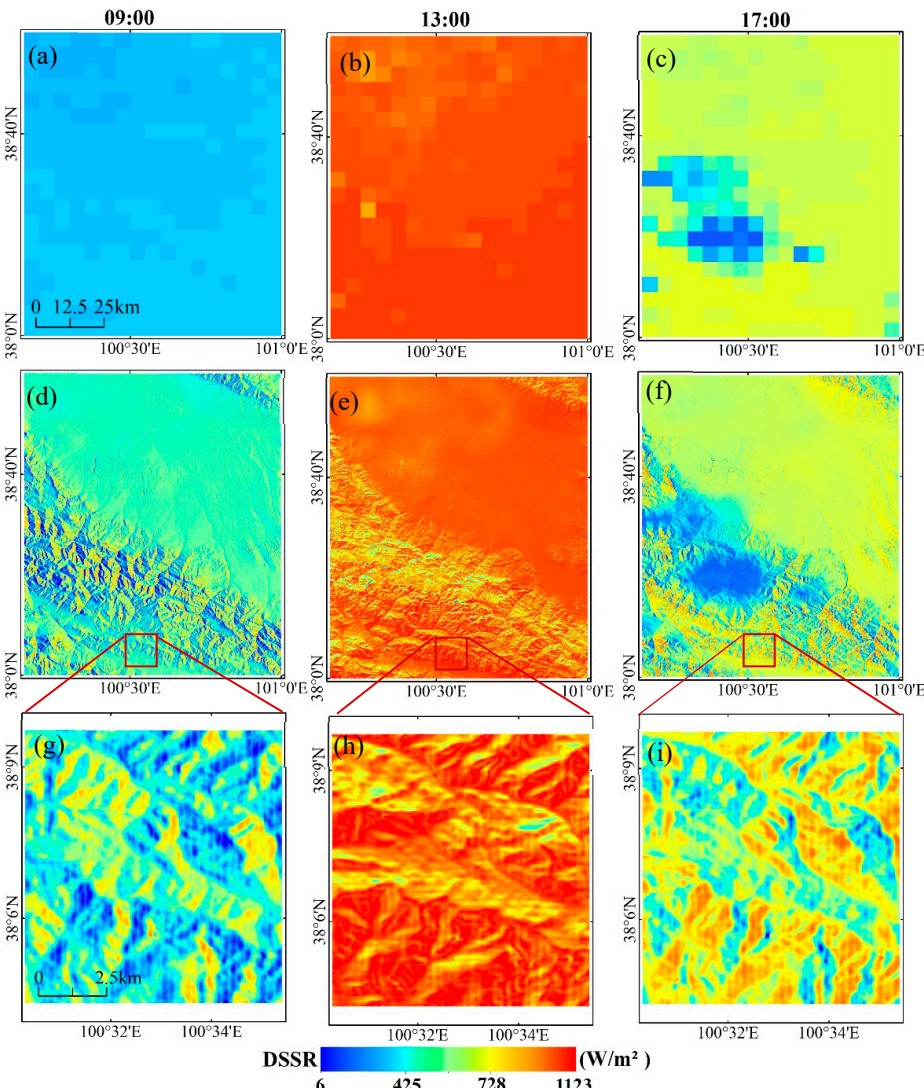

**Figure 6.** The DSSR spatial distribution at three typical moments (9:00, 13:00, and 17:00 h) on 25 July 2018: (**a**–**c**) are the original H8 radiation products, (**d**–**f**) are the downscaled DSSR data, and (**g**–**i**) show a zoomed-in view of the subregion.

To compare the spatiotemporal variation in the downscaled daily average DSSR from 2018 to 2019, eight representative periods of solar irradiance were investigated, which were basically cloud-free throughout the day in different seasons except for 10 April 2019 (blue area), as shown in Figure 7. It can be seen that the DSSR spatial distribution in a given season in different years was similar. In short, the value of shortwave solar radiation is the largest in summer (25 July 2018, and 14 August 2019), followed by spring (6 April 2018, and 10 April 2019) and fall (12 October 2018, and 26 September 2019), and the smallest in winter (22 January 2018, and 15 February 2019). However, the local topography seriously affects the seasonal variation of solar radiation. For example, on the south and southwest slopes in areas of rugged terrain, the largest DSSR values in winter (22 January 2018) are higher than those of other areas in summer (25 July 2018).

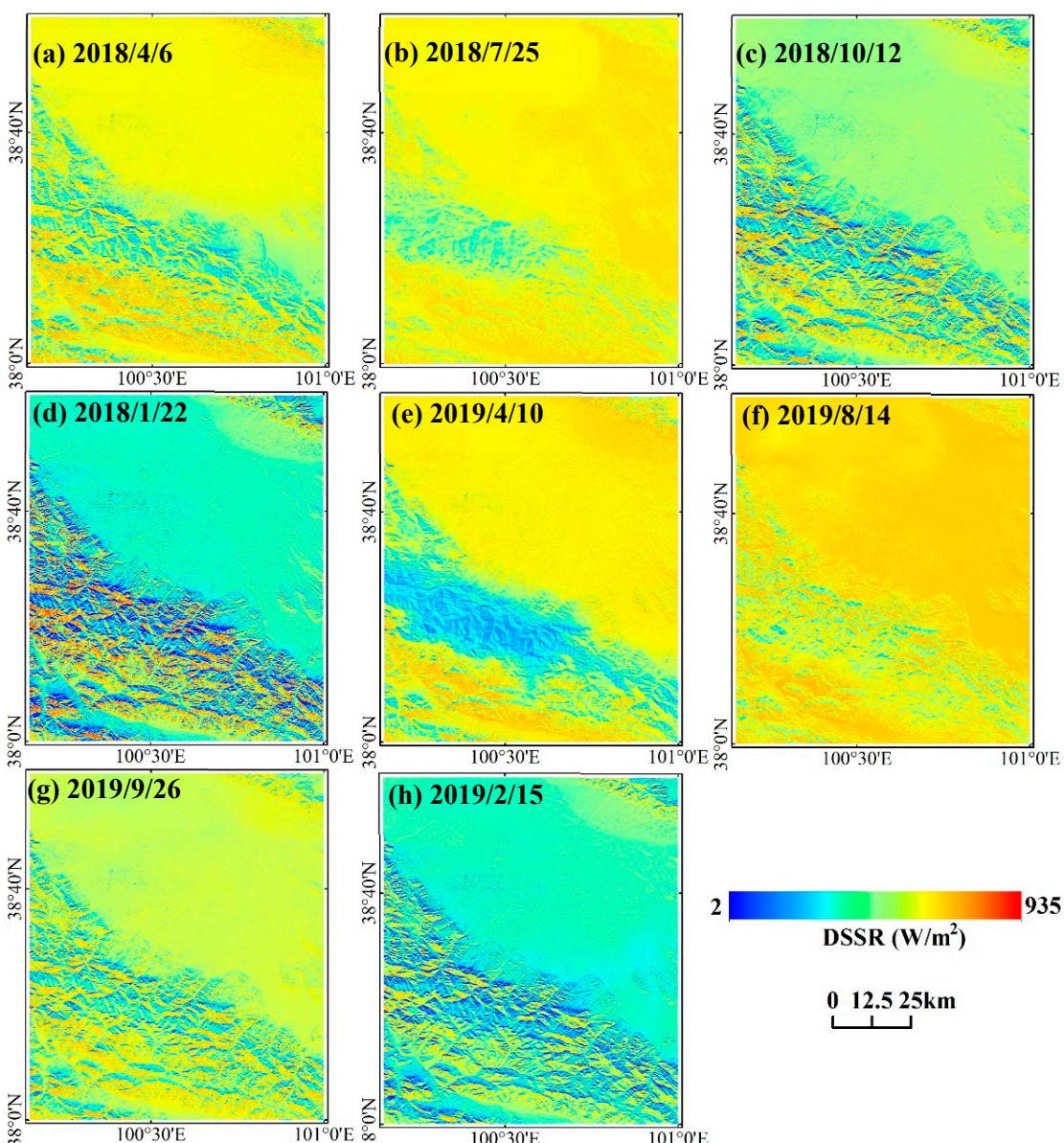

**Figure 7.** Spatial variations of daily average DSSR over 8 representative periods from 2018 to 2019.

4.2.2. Mapping of the Temporal Extrapolation DSSR

In this study, the sinusoidal model was used to estimate the daily average DSSR over 62 days during a mass-balance year from September 2017 to August 2018. Figure 8 depicts the spatial distribution characteristics of the daily average DSSR over ten typical periods in different months. It is easy to see that the value of the daily average DSSR not only has strong seasonal variation characteristics, but is also greatly affected by the local topography. Because the Laohugou area has a high altitude and complex terrain, the shortwave solar radiation received on a slope grid is affected by various terrain factors, the most influential of which is the obstruction coefficient and aspect [34].

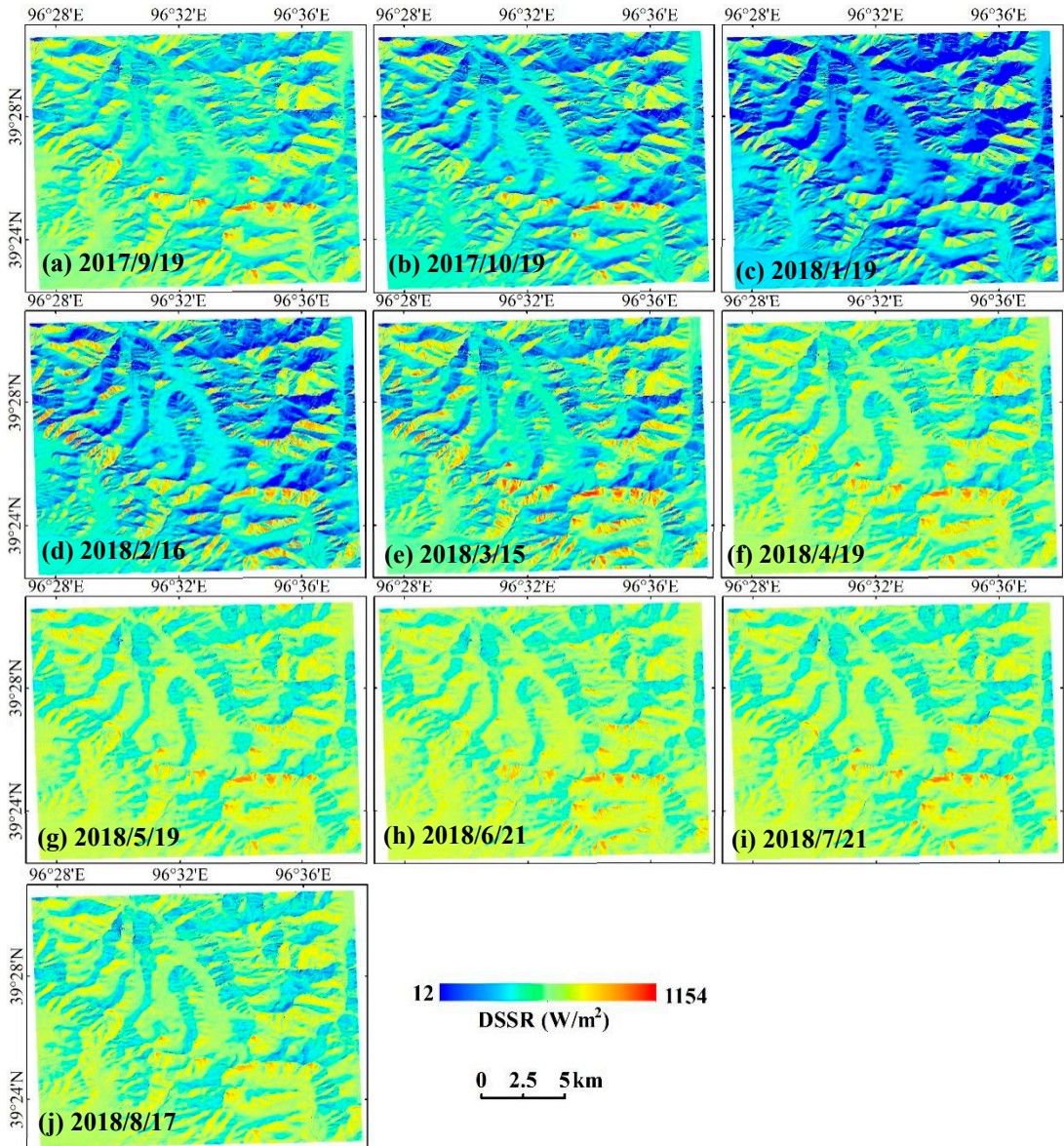

**Figure 8.** Spatial distribution of the daily average DSSR over ten typical periods during a mass-balance year from September 2017 to August 2018.

To quantify the spatiotemporal variation for the daily average DSSR in complex terrain areas, five representative topographic locations, i.e., P1, P2, P3, P4, P5 and the regional average (averaged DSSR over the whole domain) for analysis. Among them, P2 and P4 are located in relatively flat areas at different elevations, as shown in Figure 9a. Point P1 is located on the 52° slope of the south, point P3 is located on the 39° slope of the northeast slope, and point P5 lies on a 46° slope in the north. Figure 9b depicts the variation curves of the daily average solar irradiance at the five points and the regional average DSSR. It is easy to see that the DSSR seasonal variation is similar to the regional average (sinusoidal) at the two points on the flat glacier surface. The solar irradiance at P3 and P5 on the north and northeast slopes was low throughout the year, even the DSSR value is close to zero from November to January of the next year. However, the solar radiation at point P1 is very high during the same period, even exceeding the solar constant ($1367 \pm 7$ W/m$^2$), which is caused by two main reasons: one is that the cosine of the solar illumination angle at the southern slope is always very high; the other is that the surrounding-reflected radiation at P1 is extremely strong because the surrounding surface is covered with snow of ice [34].

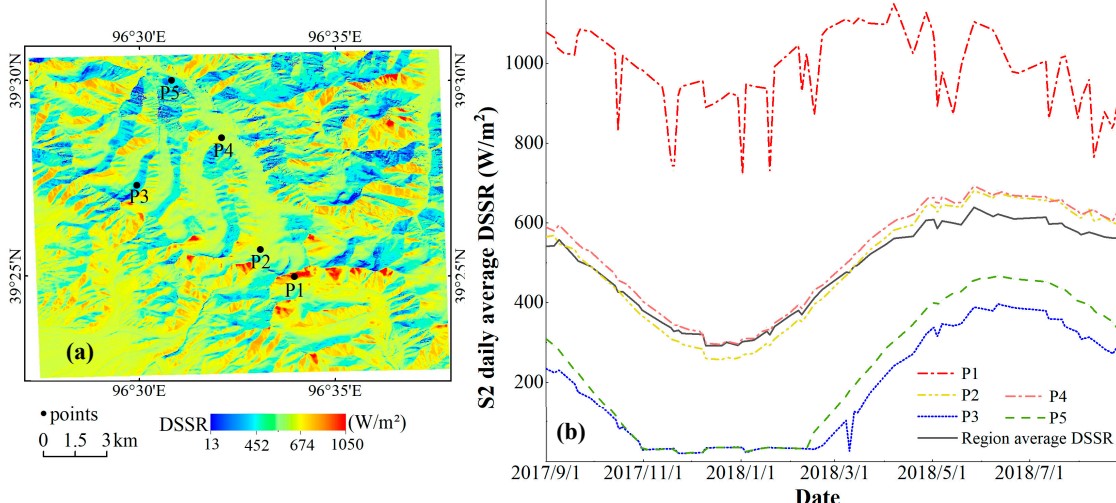

**Figure 9.** Variation trend for the daily average DSSR from September 2017 to August 2018: (**a**) locations of the five points on the DSSR map and (**b**) temporal variation curves of solar radiation.

### 4.3. Comparison of the Daily Average DSSR Estimated by Two Methods

To obtain high-resolution DSSR on a daily scale, the spatial downscaling method based on the geostationary satellite and the temporal extrapolation method based on the polar-orbiting satellite methods were explored in the UM-HRB and LHG, respectively. Therefore, it is necessary to compare the two methods in the same study area, and the LHG area was selected for a comparative analysis considering the convenience of existing datasets.

Figure 10 shows the temporal and spatial distributions of solar radiation for the 10-min DSSR estimated by the sinusoidal model and spatial downscaling at different moments on 17 August 2018. Due to the length of the paper, only 14 full hours of DSSR, from 7 to 20:00 h are displayed. From the perspective of visual effects, the DSSR spatial distribution of the two methods was found to be similar for a given moment of the day. The value of the DSSR increases gradually from sunrise to noon but decreases gradually from noon to sunset. The east slope receives more solar radiation in the morning, while the west slope receives more solar radiation in the afternoon. However, it is worth noting that compared with the 50 m H8 DSSR, the 10 m S2 DSSR shows two prominent differences under different terrain conditions and different moments of the daytime. Firstly, the sunny slope DSSR appears relatively strong during the period when the solar radiation value is high (such as 11:00–16:00) due to the consideration of the surrounding-reflected radiation component from the surrounding terrain. Secondly, in the area with a small solar radiation value (such as 18:00), the DSSR heterogeneity obtained by the temporal extrapolation method is smaller than that obtained by the spatial downscaling method because terrain factors, such as the cosine of the solar illumination angle at the current moment, are not recalculated.

In order to further quantitatively evaluate the estimation accuracy of the two methods, 28 ground-based measurements of station AWS2, collected from 7:00 to 20:30 h on 17 August 2018 were selected for variation. Since ground measurements in the LHG are recorded every 30 min, three 10-min DSSR values estimated by the two algorithms mentioned above were aggregated to obtain the spatial and temporal distributions of the 30 min averaged data. As shown in Table 4, the estimated values obtained with the two methods are in good agreement with those of the ground observers, and the $R^2$ value is greater than 0.96. The results reveal that the accuracy of the downscaled 50 m H8 DSSR is higher than that of the temporal extrapolation. It is unclear whether the 10 m or the 50 m DSSR estimated by the temporal extrapolation method has a larger statistical dispersion, as they have RMSE values of 94.77 W/m$^2$ and 95.97 W/m$^2$, respectively.

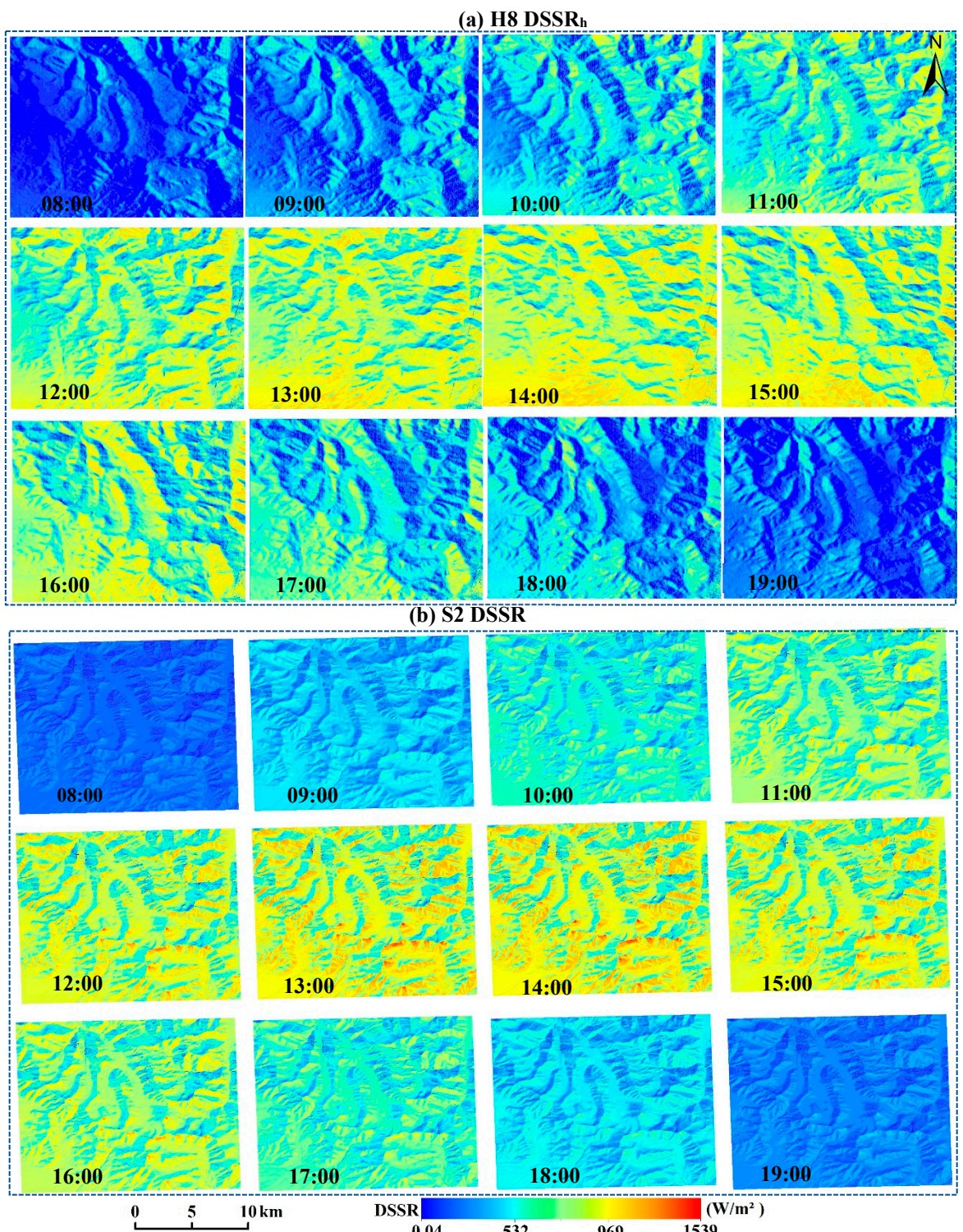

**Figure 10.** Spatial distribution of the H8 spatial downscaled DSSR and the S2 DSSR estimated by the sinusoidal model for 17 August 2018: (**a**) H8 DSSR$_h$ and (**b**) S2 DSSR.

In addition, the spatial heterogeneity of 50 m daily average DSSR data downscaled by H8 products on 17 August 2018 was lower than 10 m temporal extrapolation DSSR obtained by the S2, as shown in Figure 11. It can be seen that the daily average DSSR of 50-m H8 and 10-m S2 are quite different in some regions, which are mainly caused by two reasons. On the one hand, the 50-m H8 DSSR has low spatial resolution and the value is spatially smoothed; on the other hand, the S2 solar radiation inversion model considering

the reflected radiation of the surrounding terrain, which is also a large value in complex terrain areas [34].

**Table 4.** Comparison of the DSSR accuracy estimated by the two methods with 28 ground-based measurements of AWS2 for August 17, 2018.

| Data | MBE (W/m²) | MBE% | RMSE (W/m²) | RMSE% | R² |
|---|---|---|---|---|---|
| H8 DSSR$_h$ (50 m) | −2.39 | −0.39 | 76.69 | 12.60 | 0.96 |
| Sinusoidal model (10 m) | 16.66 | 5.09 | 94.77 | 15.41 | 0.97 |
| Sinusoidal model (50 m) | 10.70 | 1.76 | 95.97 | 15.77 | 0.97 |

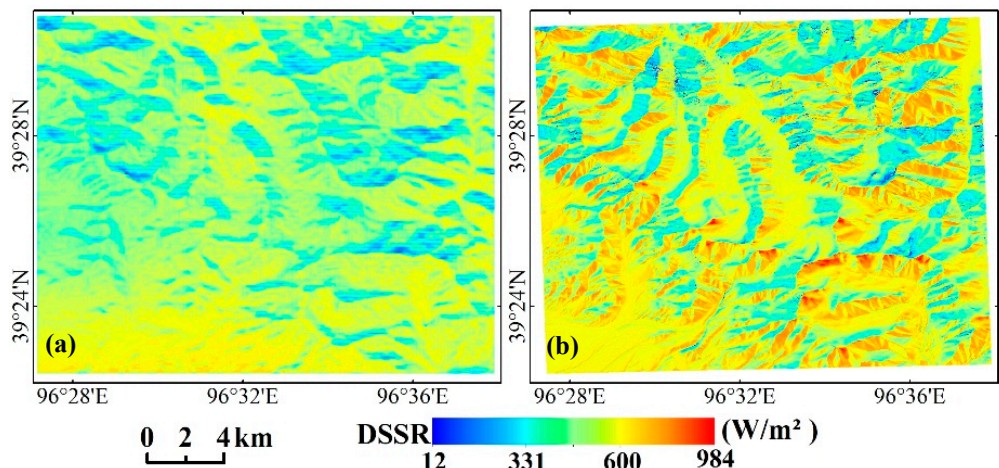

**Figure 11.** Spatial distribution of the daily average DSSR: (**a**) downscaled DSSR by H8, (**b**) the 10 m daily average DSSR data estimated by the sinusoidal model, respectively.

## 5. Discussion

### 5.1. Uncertainty Analysis of the Spatial Downscaling

The accuracy of spatial downscaling of DSSR based on the H8 radiation product is generally high; however, statistical results revealed that its reliability varies among observation sites in the UM-HRB, as shown in Table 5. Except for the DYK station, the correlation between the spatially downscaled DSSR and ground measurements was high, and the R² values were greater than 0.97. The MBE was greater than zero at different sites, indicating that the spatial downscaling results overestimated the ground shortwave solar radiation in the region. Among the six stations, the statistical dispersion of the HRS station was the smallest (RMSE = 42.89 W/m²) and that of the DYK station was the largest (RMSE = 102.91 W/m2). In general, the spatial downscaling of DSSR data was the most accurate at the HRS station, and the overall accuracy at the DYK station was the lowest, which is closely related to the spatial representation of the measurements at the station. Further research found that the reasons for the largest RMSE and MBE of the DSSR spatial downscaling method mainly come from three aspects: the accuracy of the original H8 radiation products; the complexity of the terrain, the more fragmented the terrain, the lower the DSSR estimation accuracy; and the local weather conditions [46].

In general, the verification accuracy of downscaled DSSR data depends not only on the locations of ground stations but also on the weather conditions. Taking 15 February 2019 as an example (Figure 12), the diurnal variation curve of downscaled DSSR is basically consistent with the observation curve, and both present a smooth sinusoidal curve in fully clear-sky weather conditions. However, at the DM station, the difference between the two solar radiation curves at noon is large due to the influence of cloudy weather. Generally speaking, the downscaled DSSR is slightly higher than the ground measurements around noon, while it is similar to the ground observations after sunrise and before sunset, except for at the DYK site due to the terrain.

**Table 5.** Accuracy verification statistics of spatially downscaled DSSR in the middle and UM-HRB at different stations.

| Station | MBE (W/m$^2$) | MBE% | RMSE (W/m$^2$) | RMSE% | R$^2$ | Number of Measurement Points |
|---|---|---|---|---|---|---|
| ZY | 38.25 | 7.48 | 55.42 | 10.83 | 0.98 | |
| HZZ | 43.36 | 8.43 | 64.81 | 12.60 | 0.97 | |
| AR | 35.17 | 6.47 | 60.16 | 11.07 | 0.98 | 531 |
| HRS | 29.73 | 5.73 | 60.16 | 11.59 | 0.99 | |
| DYK | 30.19 | 5.85 | 102.91 | 19.93 | 0.89 | 409 |
| DM | 69.93 | 14.29 | 82.53 | 16.86 | 0.98 | 469 |

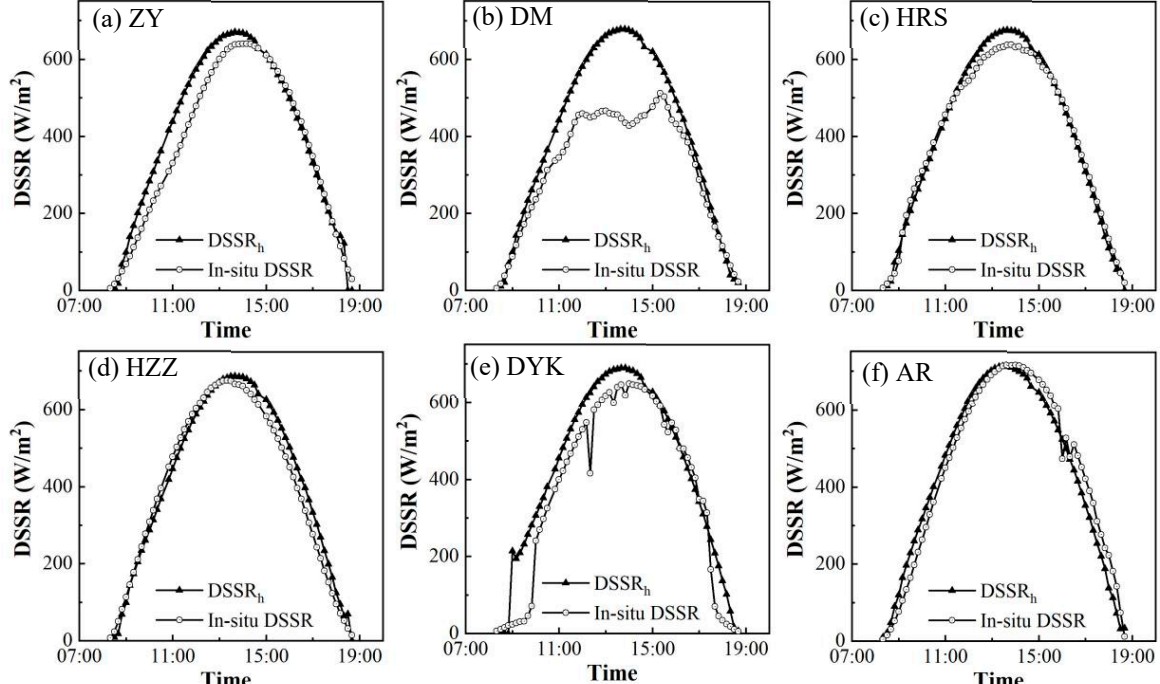

**Figure 12.** The diurnal cycle of DSSR from downscaled H8 and ground observations after cosine correction at six sites on 15 February 2019: (**a**) ZY; (**b**) DM; (**c**) HRS; (**d**) HZZ; (**e**) DYK; (**f**) AR.

## 5.2. Uncertainty Analysis of the Temporal Extrapolation

The sinusoidal expansion model can estimate the daily average solar radiation based on high-precision DSSR at the S2 satellite transit time. However, the reliability of using the sinusoidal model to simulate instantaneous solar radiation at a certain moment during the daytime needs further analysis. Figure 13 illustrates the values estimated by the sinusoidal model and the ground-observed DSSR daytime change curve at the station AWS2 on 17 August 2018. The estimation curve is basically consistent with the variation curves of the field measurements. After a careful analysis, we found that the estimated DSSR values near the periods of sunrise and sunset were higher than the ground measurements, while the values collected around noon were lower than the observations. In addition, the DSSR curve of the ground measurements was concave at around 16:00 due to the short-lived clouds, and the estimated value was higher than the ground measurement value during this period.

The sinusoidal model assumes that DSSR variation follows a sinusoidal distribution within a day [17,18] and ignores the fluctuations in the individual daytime solar radiation curve caused by weather and other reasons. This will cause high estimates to be greater than ground-measured values [47]. Furthermore, the temporal extrapolation method estimates the DSSR distribution throughout the day based on the shortwave solar radiation of a

single satellite transit time, especially at noon, which may cause large uncertainty in DSSR values estimated close to sunrise and sunset due to the neglect of the influence of the solar illumination angle. Yan et al. [23] found that under different terrain conditions, the satellite overpass times at 10:00 and 15:00 h are most suitable for daily extrapolation. Overall, the uncertainty of the DSSR sinusoidal temporal extrapolation method consists of three key reasons: the DSSR estimation accuracy at the S2 satellite transit time; the influence of weather conditions during the daytime, especially the occurrence of transient clouds; and the influence of topographic factors related to local moments, such as the cosine of the solar illumination angle.

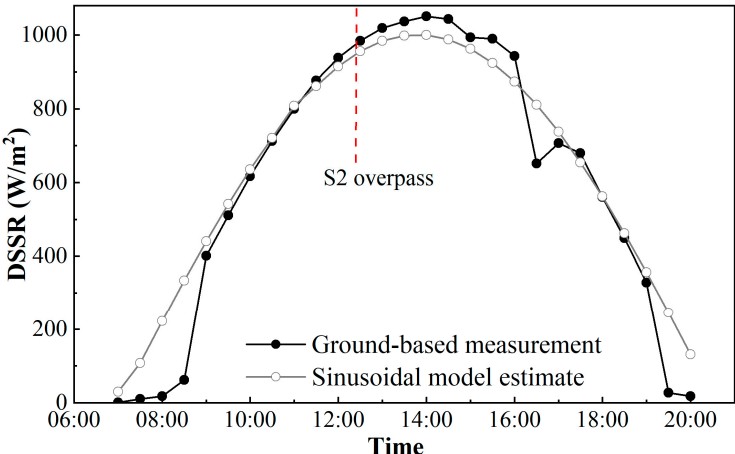

**Figure 13.** Comparison of diurnal variation in shortwave solar radiation data obtained with the sinusoidal model and ground measurements on 7 August 2018.

## 6. Conclusions

This study made full use of the respective advantages of geostationary satellites with the high temporal resolution and high spatial resolution of polar-orbiting satellites and determined the daily average solar radiation based on the spatial downscaling method and the temporal extrapolation method in a mountainous area. The former takes the 10-min 5 km radiation products as input, considers local terrain factors such as terrain shading, obtains continuous downscaled products with a 50 m resolution, and then integrates those to obtain the daily average DSSR. The latter uses the instantaneous 10 m instantaneous solar radiation estimated by S2 satellites based on a mountain radiation scheme as the input and uses the empirical sinusoidal model to obtain the daily average DSSR. The verification results confirm that both spatial downscaling and temporal extrapolation can provide reliable daily average DSSR data for mountainous areas without relying on in situ observations; thus, they can be used to provide basic data for local regional ecological, hydrological, and glacier simulation research.

However, both methods of estimating average daily DSSR have some shortcomings. First, in the spatial downscaling method, a simple parameterized empirical formula is used due to the limitation of obtaining high-resolution atmospheric parameters, and the surrounding reflected irradiance contributed from observed pixels is ignored. Second, in the temporal extrapolation method, in temporal extrapolation, the sunrise and sunset are calculated based on the date and the latitude and longitude of the study area without considering the influence of the local terrain. Third, the two methods mentioned above are more suitable for clear-sky days, without considering the influence of clouds. In the future, we should improve the two methods by introducing atmospheric products such as clouds, aerosols, and water vapor with higher spatial and temporal resolutions, which can be applied to all-sky conditions. On the other hand, we should integrate the two methods to truly combine high-time resolution geostationary satellite products with high-spatial-resolution polar-orbiting satellites to estimate accurately the daily average DSSR.

**Author Contributions:** Y.Z. designed the research, edited and analyzed the paper. L.C. formulated the model, prepared the original draft, and processed the satellite products and the site observations. All authors have read and agreed to the published version of the manuscript.

**Funding:** This research was funded by the National Natural Science Foundation of China (NSFC) project under grant numbers 41871277 and 41561080.

**Acknowledgments:** The ground observation radiation datasets were provided by the National Tibetan Plateau Data Center (http://data.tpdc.ac.cn/zh-hans/, (accessed on 1 June 2022)) and the Qilian Shan Station of Glaciology and Ecological Environment. The Himawari-8 10 min radiation products and Sentinel-2 and DEM data used in this article were obtained from the Japan Aerospace Exploration Agency (JAXA, https://www.eorc.jaxa.jp/ptree/index.html, (accessed on 1 June 2022)), ESA Copernicus Open Access Hub (https://scihub.copernicus.eu/, (accessed on 1 June 2022)) and the National Tibetan Plateau Data Center, respectively.

**Conflicts of Interest:** The authors declare no conflict of interest.

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
