# Peer review of "Estimation of Daily Average Shortwave Solar Radiation under Clear-Sky Conditions by the Spatial Downscaling and Temporal Extrapolation of Satellite Products in Mountainous Areas"

_remotesensing, doi:10.3390/rs14112710_

Round 1

Reviewer 1 Report

This manuscript presents a technique for obtaining daily average downward surface shortwave radiation using the spatial downscaling and temporal extrapolation methods. The verification results as described in the manuscript confirm that both spatial downscaling and temporal extrapolation can be employed to provide the reliable daily average downward surface shortwave radiation for the mountainous areas without relying on in situ observations. 

The work presented in this manuscript is interesting and a significant contribution to the field and the community. Furthermore, this manuscript is well written and credibly conveys the findings. Therefore, I have no hesitation in recommending this manuscript to be accepted for publication.

Author Response

Thank you very much for reviewing the paper. Your affirmation is a strong driving force for us to continue our in-depth research and active exploration. Thank you!

Reviewer 2 Report

Review of "Estimation of Daily Average Shortwave Solar Radiation by the Spatial Downscaling and Temporal Extrapolation of Satellite Products in Mountainous Areas" by Yanli Zhang and Linhong Chen.

The presented work certainly is of interest for a community working with local daily cycles of irradiance, e.g. biologist or solar energy. It seems to be a logical intermediate step using methods from literature to eventually combine polar orbiting satellite (high spatial resolution, once/twice a day) and geostationary satellite data (low spatial resolution, high temporal). The authors have worked on this topic before and part of the referenced literature is their own. Unfortunately this leads to my major point of critique. The manuscript skips many details and does not put enough effort into the presentation of the used methods basics. The results are quite interesting and worth publication, but the manuscript misses out on providing proper understandable estimates of quality of the presented results or improvement in comparison to earlier steps.

That mean, I mainly see a problem in quality of presentation on several levels. Most individual points are of minor importance, but the sum of issues unfortunately makes it difficult for the reader to follow.

I recommend major revision of the presentation before publication.

Major points: 

Already the manuscript title, but also the abstract, should contain a statement that the manuscript is only valid for "clear-sky"!

The introduction is missing a description of the user community and whether they need what the manuscript offers. 

The methods section suffers from incomplete descriptions. Often details are left out or a reference is given with no facts formulated. The reader finds out most of the things while reading through the rest of the manuscript (many pages later), but it would be much better to give more complete descriptions early. Short details (often a half sentence or one sentence) on used methods and data sets would improve readability in many places. 

The results section includes a few lengthy descriptions of  simple and obvious things in the images, but does not comment on the interesting deviations from the expected.

The assessment of the methods' qualities is weak. Most of the time statements like "good", "precise", "consistent" are given without any proof of quality, or comparison to base values (e.g. by use of relative values instead of absolute), or demonstration of improvement. This already starts in the abstract were relative MBE, RMSE could be given in addition to the absolute numbers. I suggest to include some number (e.g. extend table 3) with the impact of the several steps of processing. Then it would be very clear how much improvement over low resolution uncorrected data is reached.

Most of the time the use of the English language is good, but still there are numerous points with small errors and miss-used words, verb forms and sometime missing verbs. A careful language check with a professional or a native speaker is needed.

Specific points:

Abstract: You have to mention that you are only talking about "clear sky" in the very beginning of the abstract. And again you have to state that more clearly in the final paragraphs of your introduction.

Line 24, line 26: This type of number in an abstract is hard to digest without comparison to a base value (average DSSR) or without a statement what would be needed! 

Line 28: Please add  "...the temporal extrapolation method of instantaneous HIGH RESOLUTION radiation products can more finely describe..." 

Introduction: It would be nice, if you could tell the reader a bit more: Who needs what? Which gap does hr low freq data that you can offer? Most of the literatur is on providing data not on its use and motivation. And you can not ask the reader to find the motvation for this manuscript in other papers.

line 54: "high spatial resolution" - Please give numbers. You should also add references for these DSSR products.

line 57: Again it would be nice to have some reference for this sentence's statements. Please also add one or two sentences what "interpolation method, MRM and sinusoidal model" mean. Similar in the next line: Which "atmospheric parameters"?

line 70-74: Please give a few more words for each cited work. This is all too short to be easily readable.

line 80: "high temporal resolution" - Again, please add numbers.

line 97: Please shortly explain the difference between "obstruction coeff" and "sky-view factor".

line 108: BSRN. Please add reference.

line 113: Please add more information on the sentinel-2 data, resolution, channels,...

line 126: Please add the fact that this will be on clear-sky only!

line 194: A proper reference is better than this website link. Please add.

Figure 2: Here it does not look as if spatial downscaling and temporal extrapolation would be both based on DEM. DEM just stands between H8 and S2 data and it looks as if S2 data would be derived from H8 (following the arrow direction). Should there be a connection at all? Please improve. PLease add spatial resolution in the lower part.

line 227-229: Please check language. "primitively downscaled" is not understandable. "each downscaled DSSR componet is seperately performed topographic correction" is an ill-constructed sentence part and "real" is not real, but only "50 m resolution considering slope and elevation", if I got it right.

line 242: There is no "overpass time" for geostationary satellites. They are geostationary.

Eq: 3,4: Why do you skip the "i" subscript for Ebh and Edh and cos(theta)? 

line 262: Please make clearer the difference between "terrain factors" which are rather "local terrain 3D geometry factors" and the "1D terrain elevation correction" applied before. I find the use of terms confusing here. Maybe add the steps to the flow chart in Fig 2 to clarify?

line 308: What is the "high accuracy" found in literature you reference. Please mention some numbers from the paper.

line 310-314: I think that this is still without interpolation or temporal extrapolation, right? Which steps were applied before reaching the mentioned quality? Please make clearer which data you are talking about . Can you comment on these numbers? Why are they good enough? Is it well known where the bias comes from? 

Table 3: What is "DSSRc"? I think it was not introduced before. Shouldn't the second line of the table 3 contain the line 314 data? You description sounds like this is the case. If not, please make the difference clearer. Why don't you include data from all processing steps in this table:
*DSSR H8 original low resolution accuracy
*after terrain correction
*after removal of local solar illumination angle effects
*after obstruction coefficient application
*after sky-view factors
This would be very instructive and would help you argumenting that things are "good" or "high quality". 

line 336/337/353/354/361: "good agreement", "precisely estimate", "well", "consistent", "acceptable". These all give only an unsupported opinion and not a result. The reader might have a very different opinion on the numbers.

line 356: "RMSE" combined with the words "overestimated" and "underestimated" does not make much sense.

Fig 6: What ist the blue spot in (c) and (f)? Please comment on this in the main text. The color bar as used in all map figures is not well readable. Please add more values between min and max in all figures.Please extend all captions with some more information on the shown method's results. 

Fig 7: What happened to April 10, 2019 compared to April 4, 2018? Clouds again? Please comment on the obvious difference in the manuscript. Should "period" not be "day"?

Fig 8: Why didn't you pick only 8 days as in Fig 6 and 7? The difference between the last 5 is hardly visible.

line 448: "daily average DSSR variation curve of 62 days". Is this for one site? Or averaged over the whole domain? 

line 450-457: This seems to be unnecessary description of the very obvious. The annual variation with cos(sza) has to be there. This might be a nice check for you, but you could leave it out of the publication.

line 469/470 and Fig 10: I do not think that there is not enpugh space to make a more direct comparison. If you remove the 7:00 and 20:00 images there is enough space for 12 times plotted next to the corresponding other 12 images. The total of 24 would easily fit on one page.

table 4: This is for one station. State this in the caption.

Figure 12: There is a big difference between H8 and both S2 products. Where does it come from? From the colors I would estimate differences to be 700 and and close to 1000 W/m2 in the other? 50% difference?! And not only local "hot spots". You have to discuss this in the text.

line 523-532: These is very repetitive given that we can read the table. Please focus on some important points and summarize! What are the reasons for the largest RMSE and MBE? That would be interesting!

Discussion chapter 4.1 and 4.2: Would you please summarize your uncertainty and not just discuss a few isolated cases. You have a bias of 5-10% of daily average DSSR and an uncertainty (RMSE) of 10-20% in 4.1, right? Please state this. Similar statements are missing in the end of 4.2.

Figure 14: Please mark the time of the satellite overpass.

line 613: What does this empirical formula replace? Please state. Did you ever tell us about this point before? If not, this should be part of the methods chapter.

line 617: "...based on the date and the latitude and longitude of the study area without considering the influence of local terrain." -- Why? Did you tell us about this fact before? This explains some things visible above in Fig 14. But even there you didn't tell the reader. 

Minor and some language points (latter are too numerous to list all):

Line 10: "slope" should probably be "sloped".

Line 12: "However..." - Please give another sentence of reason. Why is it difficult?

Line 16: "The upper..." Long hardly readable sentence. Please revise.

Line 21: "3002" ?

Line 43: "...especially the estimated DSSR has spatial distribution characteristics." What do you mean? Unclear.

line 107: "Link" -> "Linke"

line 116: "between Sept 2017 and August". Year is missing. Probably August 2018? 

line 199: "published ... through" There is something missing here.

line 206: "DEM ... are applied for free ...". This is probably not the right way to say "publicly available"? Or do you mean something else?

line 233: "surrounding-reflected" sounds awkward. Better "surface reflected diffuse"?

line 235/245: incorrect use of "which", misrelation to main sentence

line 253: "Rayleigh".

line 259: "...irradiance corrected by the initial downscaled DSSR with the high-resolution..." Unclear, maybe du to language.

line 287: "...using the temporal extrapolation of sinusoidal model." Language!

line 309: "valide the accuracy" Language!

line 358: "are fewer completely clear skies throughout the day, which makes the sinusoidal model with certain uncertainty." Language.

line 392: No complete sentence.

line 408: What is a mass-balance year?

line 497: Change to "of station AWS2 (compare Fig1)".

line 514: "...relationship between surface solar radiance and local terrain cannot be expressed objectively, shown as in Figure 12 (a)." Please clarify in the text. I do not understand.

Table 5: The number "531" in the last column is valid for the first for. At the moment this is unclear.

line 550: "Furthermore, the downscaled DSSR is ..." In this sentence you are  not talking about DM anymore. Unclear.

line 578: What is the "4550 station"?

line 590: "...and will cause high estimates to be greater than ground-measured values [46]." - Unclear due to language issues.

Author Response

We sincerely thank your detailed, valuable, and insightful comments.We have redrawn almost all figures and tables, and have provided point-by-point responses below. The comments have been copied below and are followed by an explanation of how they were addressed in the resubmitted manuscript. Your original comments appear in black; our responses follow in blue.

The modifications are marked with red text in the paper.

Round 2

Reviewer 2 Report

Dear Authors,

I do see clear improvements. Thank you for that!

Still in some points I do not understand your replies to my comments, although you indicated agreement. Would you please check my remaining questions. I just use my original comments and your reply as marker.

12. line 57: Again it would be nice to have some reference for this sentence's statements. Please also add one or two sentences what "interpolation method, MRM and sinusoidal model" mean. Similar in the next line: Which "atmospheric parameters"?

Reply: After this sentence, we had explained the three methods and give the corresponding references: the interpolation method corresponding to literature [12-13], the MRM corresponding to literature [14-15], and the sinusoidal model corresponding to literature [16].
Thanks for your suggestion, we have explained the atmospheric parameters:

Excuse me, this is only a small point, but why not improve it? In your new lines 67+68 you mention three approaches. Can you please make sure that the next sentences contain three explanations making clear which is which? "The first one ..." . "The second ..." . "The third ..."

26. line 310-314: I think that this is still without interpolation or temporal extrapolation, right? Which steps were applied before reaching the mentioned quality? Please make clearer which data you are talking about . Can you comment on these numbers? Why are they good enough? Is it well known where the bias comes from?

Reply: I'm really sorry, we didn't write it clearly. This part is mainly for the accuracy verification of the original H8 product without interpolation or temporal extrapolation, so we added the words “the original H8 radiation products” in many parts in the 3.1.1 section.

Now you make clear that it is the original H8 product, thanks. Still I would, at least, like to read a statement in the text on my question "Is it well known where the bias comes from?". Please check again.

33. line 448: "daily average DSSR variation curve of 62 days". Is this for one site? Or averaged over the whole domain?

Reply: Yes, the daily average DSSR variation curve of 62 days is the average DSSR of the whole domain.

Please add that information to the text!

35. line 469/470 and Fig 10: I do not think that there is not enpugh space to make a more direct comparison. If you remove the 7:00 and 20:00 images there is enough space for 12 times plotted next to the corresponding other 12 images. The total of 24 would easily fit on one page.

Reply: Thank you for your suggestion. For the convenience of analysis and comparison, we have combined Figure 10 and Figure 11 into one figure (Figure 10).

Thank you. Because the comparison is much easier now, it becomes more striking that there are systematic differences you have to comment on in the text. Especially in the central upper half of the 1800 images these become clear! There the contrast between different slopes inclined towards and away from the sun is even opposite in the two data sets?! How could that happen? Please comment and discuss!

38. line 523-532: These is very repetitive given that we can read the table. Please focus on some important points and summarize! What are the reasons for the largest RMSE and MBE? That would be interesting!

Reply: We have added some explanations.

Where? I can just see language editing in this section?! Please state something on "What are the reasons for the largest RMSE and MBE?" in the text.

39. Discussion chapter 4.1 and 4.2: Would you please summarize your uncertainty and not just discuss a few isolated cases. You have a bias of 5-10% of daily average DSSR and an uncertainty (RMSE) of 10-20% in 4.1, right? Please state this. Similar statements are missing in the end of 4.2.

Reply: Based on your suggestion we have added explanations. I'm very sorry, the current part of the discussion may need to be further improved. In the future, we will think deeply and improve the ability of discussion and writing.

I cannot find any explanations? Maybe I was hard to understand, sorry. Can you give some assessment of your results? Is the overall uncertainty you found an improvement over earlier possibilities to retrieve daytime DSSR? Which factors contribute most to the uncertainty?

41. line 613: What does this empirical formula replace? Please state. Did you ever tell us about this point before? If not, this should be part of the methods chapter.

Reply: According to your suggestion, we have supplemented this point before.

Maybe I simply do not understand what formula you are talking about. Sorry. Where did you add a comment on this question?

Author Response

Reviewer 2’s Comments and Our Responses

Thank you very much for your careful review this paper and your very insightful comments. I'm very sorry for not answering your seven questions clearly. We have carefully revised and answered your questions one by one, and hope that the current revisions can satisfy the requirements. Your comments appear in black; our responses follow in blue.

The modifications are marked with red text in the paper.

  1. Excuse me, this is only a small point, but why not improve it? In your new lines 67+68 you mention three approaches. Can you please make sure that the next sentences contain three explanations making clear which is which? "The first one ..." . "The second ..." . "The third ..."

Reply: I'm very sorry, I just explained it to you and forgot to add the corresponding references in the paper. The three methods of temporal extrapolation and the corresponding references should now correspond to avoid confusion.

Lines 67-68:

Consequently, numerous studies have made great efforts to extend the estimated instantaneous solar irradiance to the daily average DSSR using three main methods: the interpolation method [12-13], the Meteorological Radiation Model (MRM) [14-15] and the sinusoidal model simulation method [16-23].

26.line 310-314: Now you make clear that it is the original H8 product, thanks. Still I would, at least, like to read a statement in the text on my question "Is it well known where the bias comes from?". Please check again.

Reply: We have added error sources of the original H8 product.

Lines 303-304:

Figure 3 shows that the values of the original H8 10-min products are consistent with the ground observations, the overall accuracy is relatively high (R2=0.95, RMSE=84.85 W/m2, and MBE=50.40 W/m2), and the bias comes mainly from clouds, aerosols and bright albedo [38].

  1. line 448: Please add that information to the text!

Reply: We have added this information to the text.

Lines 408-409:

To quantify the spatiotemporal variation for the daily average DSSR in complex terrain areas, five representative topographic locations, i.e., P1, P2, P3, P4, P5 and the regional average (averaged DSSR over the whole domain) for analysis.

  1. line 469/470 and Fig 10:Thank you. Because the comparison is much easier now, it becomes more striking that there are systematic differences you have to comment on in the text. Especially in the central upper half of the 1800 images these become clear! There the contrast between different slopes inclined towards and away from the sun is even opposite in the two data sets?! How could that happen? Please comment and discuss!

Reply: Thanks for your suggestion, we have added comment and discuss.

Lines 442-450:

However, it is worth noting that compared with the 50 m H8 DSSR, the 10 m S2 DSSR shows two prominent differences under different terrain conditions and different moments of the daytime. Firstly, the sunny slope DSSR appears relatively strong during the period when the solar radiation value is high (such as 11:00-16:00) due to the consideration of the surrounding-reflected radiation component from the surrounding terrain. Secondly, in the area with small solar radiation value (such as 18:00), the DSSR heterogeneity obtained by the temporal extrapolation method is smaller than that obtained by the spatial downscaling method because terrain factors, such as the cosine of the solar illumination angle at the current moment, are not recalculated.

  1. line 523-532: Where? I can just see language editing in this section?! Please state something on "What are the reasons for the largest RMSE and MBE?" in the text.
    Reply: Thanks for your suggestion, we have the reasons.

Lines 495-498:

Further research found that the reasons for the largest RMSE and MBE of the DSSR spatial downscaling method mainly comes from three aspects: the accuracy of the original H8 radiation products; the complexity of the terrain, the more fragmented the terrain, the lower the DSSR estimation accuracy; and the local weather conditions [46].

  1. Discussion chapter 4.1 and 4.2:

I cannot find any explanations? Maybe I was hard to understand, sorry. Can you give some assessment of your results? Is the overall uncertainty you found an improvement over earlier possibilities to retrieve daytime DSSR? Which factors contribute most to the uncertainty?

Reply: We have analyzed the three main factors of uncertaintythe reasons.

Lines 495-498:

Overall, the uncertainty of the DSSR sinusoidal temporal extrapolation method consists in three key reasons: the DSSR estimation accuracy at the S2 satellite transit time; the influence of weather conditions during the daytime, especially the occurrence of transient clouds; and the influence of topographic factors related to local moments, such as the cosine of the solar illumination angle.

  1. line 613:

Maybe I simply do not understand what formula you are talking about. Sorry. Where did you add a comment on this question?

Reply: Sorry. We have added the algorithm description to the previous text of the paper.

Line 237:

the direct and diffuse transmittance of the i-th pixel and can be calculated according to the simple parameterized empirical formula presented by Wang et al. [28] by considering the Linke turbidity factor, relative optical air mass, Rayleigh optical thickness, and surface elevation.
